# Altered Expression of AQP1 and AQP4 in Brain Barriers and Cerebrospinal Fluid May Affect Cerebral Water Balance during Chronic Hypertension

**DOI:** 10.3390/ijms232012277

**Published:** 2022-10-14

**Authors:** Ibrahim González-Marrero, Luis G. Hernández-Abad, Miriam González-Gómez, María Soto-Viera, Emilia M. Carmona-Calero, Leandro Castañeyra-Ruiz, Agustín Castañeyra-Perdomo

**Affiliations:** 1Departamento de Ciencias Médicas Básicas, Facultad de Ciencias de la Salud, Campus de Ofra, Universidad de La Laguna (ULL), 38200 Santa Cruz de Tenerife, Spain; 2Instituto Universitario de Neurociencia (IUNE), Campus de Guajara, Universidad de La Laguna, 38205 Santa Cruz de Tenerife, Spain; 3Instituto de Tecnologías Biomédicas de Canarias, Universidad de La Laguna, 38071 Santa Cruz de Tenerife, Spain; 4Instituto de Investigación Puerto del Rosario, 35600 Las Palmas de Gran Canaria, Spain; 5CHOC Children’s Research Institute, 1201 W. La Veta Avenue, Orange, CA 92868, USA

**Keywords:** aquaporins, hypertension, SHR, hydrocephalus, choroid plexus, cerebrospinal fluid, ependyma

## Abstract

Hypertension is the leading cause of cardiovascular affection and premature death worldwide. The spontaneously hypertensive rat (SHR) is the most common animal model of hypertension, which is characterized by secondary ventricular dilation and hydrocephalus. Aquaporin (AQP) 1 and 4 are the main water channels responsible for the brain’s water balance. The present study focuses on defining the expression of AQPs through the time course of the development of spontaneous chronic hypertension. We performed immunofluorescence and ELISA to examine brain AQPs from 10 SHR, and 10 Wistar–Kyoto (WKY) rats studied at 6 and 12 months old. There was a significant decrease in AQP1 in the choroid plexus of the SHR-12-months group compared with the age-matched control (*p* < 0.05). In the ependyma, AQP4 was significantly decreased only in the SHR-12-months group compared with the control or SHR-6-months groups (*p* < 0.05). Per contra, AQP4 increased in astrocytes end-feet of 6 months and 12 months SHR rats (*p* < 0.05). CSF AQP detection was higher in the SHR-12-months group than in the age-matched control group. CSF findings were confirmed by Western blot. In SHR, ependymal and choroidal AQPs decreased over time, while CSF AQPs levels increased. In turn, astrocytes AQP4 increased in SHR rats. These AQP alterations may underlie hypertensive-dependent ventriculomegaly.

## 1. Introduction

The regulation of water balance in the brain is fundamental. A disturbance in this balance results in excess brain water content that contributes significantly to the pathophysiology of traumatic brain injury, hydrocephalus, and a range of neurological disorders [1,2,3]. Although about 80% of the brain is water, there is relatively little knowledge about the physiology of brain water. The water transport is mediated by aquaporin channels (AQP), a family of membrane proteins that allow passive bidirectional transport of water in response to hydrostatic and osmotic pressure [4,5]. AQP1 and AQP4 are expressed in different components of the brain barriers, such as the blood–brain barrier (BBB), the blood–cerebrospinal fluid barrier (BCSFB), and the interfaces between the brain and the brain-cerebrospinal fluid interface (brain–CSF interface), which provide a stable microenvironment that is critical for brain function [6].

The regulation of water balance in the brain is fundamental. A disturbance in this balance results in excess brain water content that contributes significantly to the pathophysiology of traumatic brain injury, hydrocephalus, and various neurological disorders [1,2,3]. Although about 80% of the brain is water, there is relatively little knowledge about the physiology of brain water. The water transport is mediated by aquaporin channels (AQP), a family of membrane proteins that allow passive bidirectional water transport in response to hydrostatic and osmotic pressure [4,5]. AQP1 and AQP4 are expressed in different components of the brain barriers, such as the blood–brain barrier (BBB), the blood–cerebrospinal fluid barrier (BCSFB), and the interfaces between the brain and the brain-cerebrospinal fluid interface (brain–CSF interface), which provides a stable microenvironment that is critical for brain function [6].

AQP1 plays an essential role in water movement through choroid plexus (ChP) epithelial cells. AQP1 is primarily located in the apical membrane of the ChP epithelium [7] and, consequently, has been implicated in playing a pivotal role in CSF secretion [8,9]. Likewise, AQP4 is expressed in glia at the borders between the main water compartments and the brain parenchyma, which include the astroglia foot processes at the BBB, the glial limiting membrane, and the ependymal and subependymal astroglia at the ventricular brain–CSF interface. This pattern of protein expression indicates the participation of AQP4 in the movement of water inside and outside the brain [10,11].

Chronic hypertension (HT) has been related to the development of brain damage, dementia, and other dysfunctions of the central nervous system, such as atherosclerotic changes or alterations of cerebral autoregulation and a progressive increase in blood flow of cerebral vessels [12,13,14,15]. In addition, according to some authors, chronic HT may cause alterations in both BCSFB and BBB [12,15,16].

The spontaneously hypertensive rat (SHR) is the most used animal model of hypertension, and the most suitable control strain is the Wistar–Kyoto (WKY) rat. It has been described as progressive ventricular dilatation in SHR between 4 and 8 weeks of age compared to the control WKY rat, but the mechanisms underlying hydrocephalus in SHRs are still uncertain [17,18,19,20]. Recent studies revealed that both male and female SHRs had hydrocephalus at week 7, which got worse at week 9 and was more severe in males [21,22]. Changes in CSF homeostasis have been heavily associated with the pathophysiology of hydrocephalus [22,23]. Alterations in AQPs expression may contribute to hydrocephalus due to impaired water reabsorption in the venous compartment [24,25,26]. Previous reports showed that the ventricular system anatomy and associated CSF transport pattern differed between SHR and WKY rats due to hydrocephalus in SHRs, cerebroventricular volumes are larger and brain volumes significantly smaller in SHR compared to WKY rats [27].

The present study focuses on the expression of AQP1 and AQP4 in the brain tissue and CSF of SHR and WKY rats to determine the time course of the development of spontaneous hydrocephalus and brain water balance along with chronic HT.

## 2. Results

### 2.1. AQP1 in ChP Epithelium (BCSFB)

Light microscopy and confocal microscopy showed the localization of AQP1 immunoreactivity in the ChP epithelium. Positive immunostaining was observed in the apical membrane of the ChP cells (Figure 1). AQP1 intensity increased significantly (*p* < 0.05) in the ChP of 6 months SHR (55 ± 5.2 relative units) compared to 6 months WKY rats (49 ± 4.7 relative units) (Figure 1a,c,e,g). However, AQP1 labeling in the ChP was reduced in the SHR-12-months group (38 ± 4.3 relative units) compared to the 12 months control group (47 ± 3.2 relative units) (Figure 1b,d,f,h). These differences were statistically significant (*p* < 0.05) (Figure 1i). Another noteworthy aspect of the research was the intragroup difference between 6 months and 12 months. While the differences in AQP1 expression were minimal in WKY-6-months and WKY-12-months, the differences were significant in the SHR (6 months 55 ± 5.2 vs. 12 months 38 ± 4.3 relative units). Therefore, the combination of HT and aging cause a drastic decrease in AQP1 values in the ChP epithelium (Figure 1j).

### 2.2. AQP4 in Ependymal Cells and Subependymal Astrocytes (Brain–CSF Interface)

Ependymal cells are known to display the AQP4 water channel in the basolateral membrane domain. As expected, AQP4 was localized in the ependymal cells and the feet of astrocytes overlying the ependyma (Figure 2). The study was performed in the lateral wall of the lateral cerebral ventricle. AQP4 immunofluorescence showed no significant differences between both 6 months groups (WKY 42 ± 4.3 vs. SHR 38 ± 4.8 relative units) (Figure 2a–d), whereas significant differences (*p* < 0.05) (Figure 2e) were found between WKY-12 months (46 ± 3.6 relative units) and SHR-12 months (35 ± 4.4 relative units) in AQP4 expression (Figure 2e–h). Between 6–12 months, AQP4 increased in the ependyma of WKY rats, while the opposite was exhibited in SHR. Thus, chronic hypertension decreases AQP4 in the ependyma, which is different from what is observed in the control group, where it increases progressively (Figure 2f).

### 2.3. AQP4 in Astrocytes (BBB)

AQP4 immunostaining was detected around brain microvessels by both methods (immunofluorescence and immunoperoxidase), confirming the localization of this water channel in astrocyte end-feet processes near blood vessels (Figure 3). An increased expression of AQP4 was found in the frontal cortex, striatum, and hippocampus at 6 months and 12 months, in SHR (6 months 32 ± 5.4; 12 months 38 ± 4.9 relative units) when compared with age-matched WKY rats (6 months 21 ± 4.1; 12 months 25 ± 3.9 relative units) (Figure 3a–h). These differences were statistically significant (*p* < 0.05) (Figure 3j). A moderate increase in AQP4 staining in the astrocyte end-feet processes resulted in a trend of aging in both groups (Figure 3k).

### 2.4. AQP1 and AQP4 in CSF

The presence of AQP1 and AQP4 in CSF was measured by immunoassay technique and Western blotting. CSF from 6 months and 12 months WKY and SHR rats were extracted at six and twelve months old (Table 1 and Figure 4). The mean concentration of AQP1 in CSF from WKY and SHR at 6 months age were similar, respectively, 1.96 ± 0.13 ng/mL and 1.64 ± 1.24 ng/mL. When comparing AQP1 values in 12 months rats, AQP1 levels of SHR were higher than those of the WKY, 2.01 ± 0.1 ng/mL in WKY vs. 3.14 ± 0.8 in SHR (Table 1). AQP1 values in CSF during aging did not vary for the normotensive group, whereas a significant increase in AQP1 levels was observed in the hypertensive rat with increasing untreated hypertension time up to 12 months of age (Figure 4a).

The mean values of AQP4 in CSF from WKY and SHR at 6 months of age were slightly higher in SHR, 8.59 ± 0.59 ng/mL and 9.78 ± 0.82 ng/mL in WKY vs. SHR. In the case of comparing the values in the 12 months rats, a remarkable increase in SHR values (18.68 ± 1.33 ng/mL) was observed in comparison to WKY (11.04 ± 0.39 ng/mL) (Table 1). AQP4 increased slightly over time in the control group, whereas in the hypertensive rat, the increase was doubled when comparing the SHR groups (Figure 4b).

Western blots were performed on the CSF of 6 months and 12 months groups of WKY and SHR (Figure 4c–e), and the differences detected in AQP1 and AQP4 concentrations showed the same tendencies as those detected by ELISA. Noteworthy is the increase in the concentration of AQP1 and AQP4 in SHR-12 months (AQP1 21.17; AQP4 30.84) compared to SHR-6 months (AQP1 9.1: AQP4: 19.06) (Figure 4d,e).

## 3. Discussion

Previous investigations have studied the effects of HT on CSF production and brain barrier damage [27,28,29]. In the present work, we added the factor of aging to chronic HT to exacerbate its effects, so we took the experimental animal up to one year of age. The new feature of this study is that in SHR-12-months, we have observed changes in the expression pattern of AQPs to match those described for other pathologies such as hydrocephalus, cranioencephalic trauma, edema, or inflammation.

CSF production has both choroidal and extrachoroidal components. It has been shown that AQP1 and AQP4 play an essential role in CSF production and homeostasis. In 6 months rats, we found a higher expression of AQP1 in SHR than in WKY. These data agree with other authors [28] and are probably related to a more significant increase in CSF secretion and faster CSF turnover compared to normotensive rats [12], although recently, another work showed that the rate of CSF production was not different between WKY and SHR [30]. Removal of AQP1 decreases osmotically driven water permeability across the ChP epithelium, which reduces CSF production and intracranial pressure [31].

In the hypertensive 12 months rat, we found a significant decrease of AQP1 in the ChP epithelium compared to the 12 months control rat. SHR develops spontaneous hydrocephalus. This AQP1 decrease is described in other types of hydrocephalus. AQP1 expression was reduced by approximately 50% in kaolin-treated wild-type mice through an endocytic recovery mechanism [32]. In Texas rats with congenital hydrocephalus, choroidal AQP1 expression was reduced early in life but normalized on postnatal day 26, just before death [33]. In humans, AQP1 expression decreases in the apical pole of the ChP epithelium under hydrocephalic conditions [34]. As discussed above, there may be increased CSF production in the SHR model relative to control at six months of age and its involvement in the development of ventriculomegaly. Over time, the down-regulation of AQP1 described could involve a mechanism by which CSF production is attempted to be decreased as an adaptive mechanism.

The brain–CSF interface plays an essential role in molecule transport balance between ventricular CSF and interstitial fluid [35]. AQP4 in ependymal cells and subependymal astrocytes is an essential factor determining CSF flow across the brain–CSF interface [36]. Mice lacking Aqp4 show sporadic hydrocephalus where there is reduced transependymal CSF flow into the brain parenchyma and reduced CSF uptake through the BBB into the blood [24,37], while others find no relationship between ventricular size and CSF dynamics [38]. Increased AQP4 expression found in human and rodent hydrocephalus [39] may be a response to facilitate CSF clearance through the transependymal pathway. AQP4 appears to play a crucial role in the edema and CSF accumulations in hydrocephalus caused by CSF extravasation through the ependyma into the interstitial fluid, which may play a protective role. In the current work, we observed no significant differences when comparing AQP4 expression in 6-month rats, whereas we found a significant decrease of AQP4 in ependymal cells and subependymal astrocytes in 12 months SHR compared to controls and 6 months SHR. Some models of hydrocephalus show increased AQP4 in the ependyma [40,41,42]. Changes in AQP4 in SHR may be more involved in preventing CSF passage from the ventricle to the brain parenchyma to avoid edema than decreasing ventricular CSF volume.

AQP4 might be involved in the removal of excess water from the brain. The altered expression level of AQP4 in brain parenchyma between SHR and WKY has been reported by several authors [27,28,29]. Chronic hypertension can lead to edema formation. After measuring brain parenchymal water content, Naessens et al. [30] results were consistent with a cytotoxic edema state in the hypertensive rat. Increased AQP4 levels could accelerate the removal of edema fluid from the brain parenchyma, and decreased AQP4 expression would delay edema formation, particularly by cytotoxic mechanisms [42]. Overexpression of AQP4 has also been reported to reduce cytotoxic edema in a hypoxia-ischemia rat model [43]. Changes in AQP4 expression could be associated with the development of cytotoxic edema, representing a first step in the pathophysiology of brain injury documented in SHR [28]. In addition, AQP4 seems to play a critical role in the glymphatic system, a flow clearance pathway, where subarachnoid CSF flow into the brain parenchyma interstitial fluid through the perivascular space of the penetrating arteries and capillaries and exit through the perivenous space. The perivascular space is surrounded by astrocyte end-feet, expressing polarized AQP4 to facilitate permeability from the periarterial space toward the brain interstitium and from there to the perivenous space, ultimately clearing metabolites and solutes [44,45]. The present study has confirmed the increased expression of AQP4 in the foot extensions of astrocytes in the brain parenchyma, which resulted in an aging tendency in both groups, with AQP4 levels being higher in SHR. As suggested above, this increase in AQP4 may serve the purpose of removing excess water from the brain parenchyma through the glymphatic pathway.

Our previous results have demonstrated the presence of AQP1 and AQP4 in CSF when studying different pathologies [26,27,34,46]. In the present study, we quantified AQP1 and AQP4 by ELISA and Western blot. CSF expression of AQP1 and AQP4 was higher in 12 months SHR compared to 6 months and age-matched WKY. In SHR, AQP1 expression in ChP decreased with aging, while CSF AQP1 levels increased. Moreover, the same pattern of inverse correlation was found in AQP4 present in ependymal cells, and CSF was shown. In summary, we found that as AQPs expression in ChP and ependyma decreased over time, we observed an increase in CSF AQPs levels (Figure 5).

Brain-derived extracellular vesicles (EVs) have been identified in CSF and blood in both healthy and pathophysiological conditions [47]. It has been proposed that the presence of AQPs in CSF may be due to their secretion by EVs [26,27,46,48,49]. Our group has reported the expression of AQP4 in the EVs released into the ventricular system associated with ventricular zone glial activation [46]. A similar mechanism may underlie the presence of AQPs in the CSF of the rats studied. Brain EVs have also been identified as a potential biomarker for injury [50,51,52]. In the brain, AQP1 and AQP4 EVs may be secreted into the CSF [53,54].

We hypothesize that the mechanism of AQPs secretion to CSF in the brain may have the same mechanism as in the kidney. AQP1 and AQP2 are primarily found in urine exosomes, produced via the endosomal pathway [55,56]. Reduced AQP2 [8] and AQP1 in collecting ducts, following vasopressin release, or in response to extracellular hypotonicity leads to an increase in AQP2- and AQP1-containing EVs in the urine [57,58]. In SHR rats, this mechanism may explain the inverse association between the levels of AQPs found in the barriers and those measured in the CSF (Figure 5).

AQP alterations seem to respond to a compensatory mechanism to reduce CSF volume, and its release into the CSF is proposed as a deactivation process of CSF production. Further investigation will be necessary to understand the role of AQP1 and AQP4 in the pathological state of SHR. The underlying mechanisms of SHR-dependent hydrocephalus may provide insights into the processes involved in adult hydrocephalus, such as those associated with intraventricular and intracerebral hemorrhage, traumatic brain injury, or idiopathic hydrocephalus.

The expression of AQPs decreased in the CSF production points (ChP and ependyma) and increased in the absorption points (neurovascular unit) in 12 months of SHR as a possible mechanism for ventriculomegaly compensation.

## 4. Materials and Methods

### 4.1. Animals

A total of 20 male rats (10 WKY and 10 SHR) at 6 and 12 months of age were used, establishing four different groups: 6-month-old WKY rats, 6-month-old SHR rats, 12-month-old WKY rats, and 12-month-old SHR rats. The 6-month-old rats were referred to as the “6 months group,” while the 12-month-old rats were referred to as the “12 months group”. The 6 months and 12 months SHR groups represent chronic phases of hypertension, respectively, which progressively develop between 6 and 24 weeks. Animals were obtained from Charles River Laboratories (Barcelona), and once at the University of La Laguna, they were housed in the same environment (25 °C, 12:12-h light-dark cycle) and allowed free access to food and water. All procedures were carried out following the Animal Research: Reporting of In Vivo Experiments (ARRIVE 2.0) guidelines and European Union guidelines for the welfare of laboratory animals (Directive 2010/63/EU). The Committee approved the use of laboratory animals of Animal Use for Research at the University of La Laguna. The number of animals used and the stress and suffering of these subjects during handling and experimentation were minimized.

All animals were weighed, and blood pressure (BP) data were taken before sacrifice (Table 2). The systolic and diastolic BP were measured by a tail-cuff method with the rats under conscious conditions using a non-invasive BP measurement system (Panlab Non-Invasive Blood Pressure System for Rodents and Dogs. Harvard Apparatus, Cambridge, UK). The differences between WKY and SHR were significant when a Student *t*-test was applied (*p* < 0.05).

### 4.2. Cerebrospinal Fluid Extraction

CSF was obtained from each group (*n* = 5 per group). Once anesthetized, they were placed in a stereotactic frame, and the CSF was removed from the cistern Magna by puncture with a 28 G needle, then centrifuged (4000× *g* for 4 min.) and stored at −80 °C. The extraction was performed according to our previous experience [14]. Due to the small volumes of rat CSF obtained (~30 µL), the CSF from every group was pooled into single samples and analyzed by triplicates through ELISA (see Section 4.4). Results were confirmed by WB (see Section 4.5).

### 4.3. Immunofluorescence and Immunohistochemistry

Rats were fixed by intracardiac perfusion with paraformaldehyde 4%, dehydrated, and embedded in paraffin under standard conditions. The brains were cut into four serial coronal sections. As previously described, the coronal sections were prepared for immunofluorescence and immunohistochemistry [59,60]. All sections were incubated overnight at 4 °C with the primary antibodies (Abnova, Taipei, China): rabbit anti-AQP4 (pab20767, 1:2000) and rabbit polyclonal anti-AQP1 (pab28892, 1:1000). For immunofluorescence the sections were incubated with the following secondary antibodies; Cyanine 3 (Cy3) dye goat anti-rabbit IgG (Invitrogen, Waltham, MA, USA, A32732, 1:500). Nuclei were stained with DAPI (4′-6′ Diamidino-2-phenylindole, dihydrochloride) (Invitrogen, D1306, 1:5000). After washing, samples were mounted in Vectashield antifade media (Vector Laboratories Inc., Newark, CA, USA) for viewing with a confocal microscope (FV1000 Olympus). For immunohistochemistry, goat anti-rabbit IgG conjugated antibody (Invitrogen, A27035, 1:300) was used as a secondary antibody. Sections were developed according to the manufacturer’s instructions for Duet StreptABC complex/HRP anti-mouse/rabbit (Dako, Carpinteria, CA, USA), and the peroxidase reaction product was visualized with 3,3′-diaminobenzidine. Sections were visualized with a LEICA DMRB microscope with a LEICA DC 300F CCD camera. The omission of incubation in the primary antibody was used as a negative control.

### 4.4. Enzyme-Linked Immunosorbent Assay (ELISA)

AQP1 and AQP4 concentrations in the CSF were measured by ELISA for human AQP1 and human AQP4 (USCN life Science Inc., Wuhan, China). The standard curve was plotted from measurements taken with the standard solution (0.25–32 ng/mL for AQP1 and 62.5–4000 pg ⁄ml for AQP4). The limit of the detectable AQP1 concentration was 0.09 ng/mL and for AQP4 concentration was 27.9 pg/mL. The assay was performed as in previous articles [61,62].

### 4.5. Western Blotting

CSF (20 μL) was included in the sample buffer (100 mM Tris–HCl pH 6.8, 4% SDS, 2% bromophenol blue, 20% glycerol, 0.5% β-mercaptoethanol), heated at 95 °C for 5 min to denature the proteins and separated by SDS-PAGE on 10% gradient gels and electroblotted onto 0.45 μm polyvinylidene difluoride membranes (PVDF; Millipore, Burlington, MA, USA) using Trans-blot Turbo (Bio-Rad, Hercules, CA, USA). The Western blot was performed according to previous articles [62]. Membranes were incubated in both primary antibodies overnight at 4 °C: rabbit polyclonal anti-AQP4 (pab20767 Abnova, 1:500) and rabbit polyclonal anti-AQP1 (pab28892, Abnova, 1:500). Anti-mouse IgG labeled with peroxidase (A31341, Thermo Fisher Scientific, Waltham, MA, USA) was used as the secondary antibody at a dilution of 1:20,000 for 1 h at room temperature. The proteins recognized were detected by luminescence using Clarity Western ECL Substrate (Bio-Rad, REF: 1705061). They were then analyzed using a ChemiDoc MP device and Image LabTM Software, version 4.0.1 (Bio-Rad). The primary antibody was omitted to validate the control method specificity.

### 4.6. Image Acquisition and Immunofluorescence-Quantification

Fluorescence intensities from images were analyzed by densitometry. Immunofluorescence slides were converted to digital images by using a Confocal Laser Scanning Microscope FV1000 (Olympus GmbH, Hamburg, Germany) as 8-bit acquisitions of color. Image analysis was conducted in Image J (v. 1.43 u, NIH, Bethesda, MD, USA). Regions of Interest (ROI) were selected, and the RGB images were subsequently split into three 8-bit grayscale images containing the red, green, and blue components. The selection of the immunostaining zone was made with the freehand tool of Image J and added to the ROI manager. The mean of the obtained values (relative units) was calculated and plotted for each mean fluorescence value of the antibodies.

### 4.7. Statistical Analysis

Data analysis was performed by SPSS software, Version 23 (SPSS Inc., Chicago, IL, USA). Measurements are reported as mean ± SD. In the statistical analysis of data, a Kolmogorov–Smirnov test was used to check data normality; all were distributed normally. A one-way ANOVA was used for comparing immunofluorescence data between WKY and SHR rats were considered statistically significant at *p* < 0.05.

## Figures and Tables

**Figure 1 ijms-23-12277-f001:**
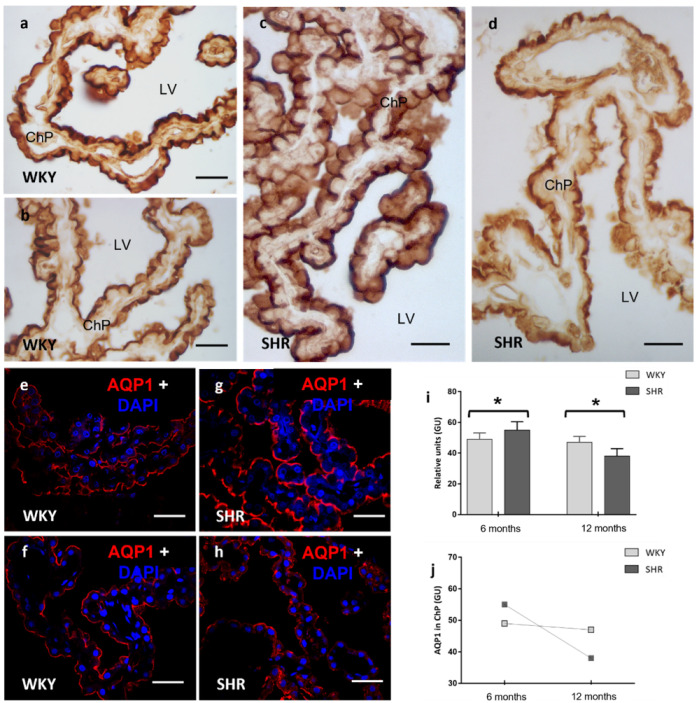
Confocal and optical microscopy images of the immunostaining of AQP1 in the ChP of WKY and SHR rats. An increase of AQP1 was observed in the ChP of SHR in the 6 months group when compared to WKY (**a**,**c**,**e**,**g**), while in the 12 months group, a decrease of AQP1 was observed in the SHR (**b**,**d**,**f**,**h**) rats. At the bottom, stain intensities values in relative units (gray units) for AQP1 (**i**,**j**) are represented as the means ± SD (*n* = five animals per group). The differences between WKY and SHR were significant. One-way ANOVA test with post hoc analysis using the Tukey post hoc test (* *p* < 0.05) was applied. LV: lateral ventricle; ChP: choroid plexus; GU: grey units. Scale bars 40 μm.

**Figure 2 ijms-23-12277-f002:**
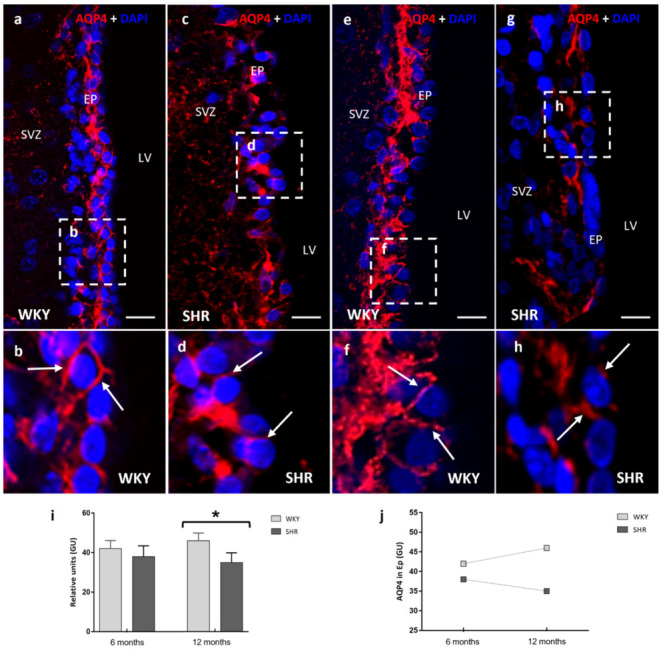
Confocal microscopy images of AQP4 in the ependymal cells of the lateral ventricle of WKY and SHR rats. A slight decrease of AQP4 was detected in the ependyma of SHR in the 6 months group when compared to WKY (**a**–**d**), while in the 12 months group, a significant decrease of AQP4 was observed in the SHR (**e**–**h**) rats. White arrows point to AQP4 immunoreactive material (**b**). At the bottom, stain intensities in relative units (gray units) for AQP4 (**i**,**j**) are represented as the means ± SD (*n* = five animals per group). The differences between WKY and SHR were significant when applied to a One-way ANOVA test with post hoc analysis using the Tukey post hoc test (* *p* < 0.05). LW: lateral wall; SVZ: subventricular zone; EP: ependyma; LV: lateral ventricle; GU: grey units. Scale bars 20 μm.

**Figure 3 ijms-23-12277-f003:**
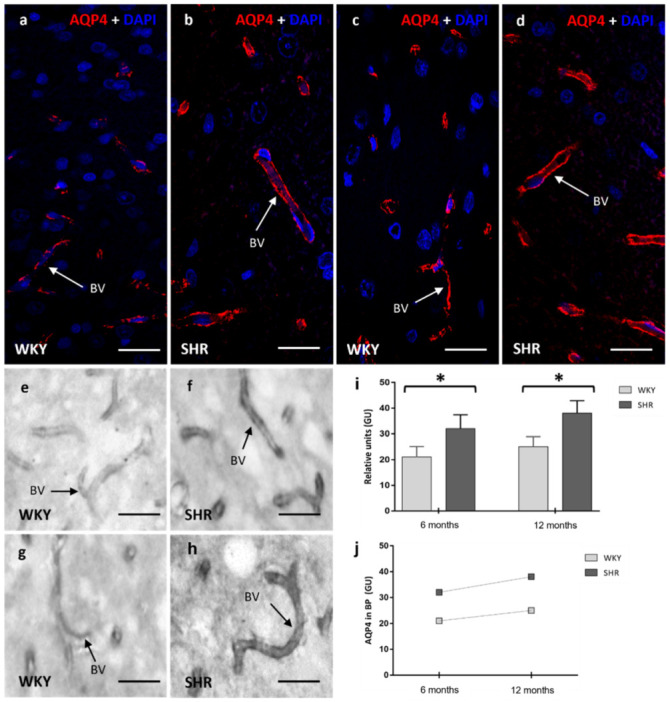
Confocal and optical microscopy images of AQP4 in the blood vessels of brain parenchyma of WKY and SHR rats. A significant increase of AQP4 was observed in SHR in the 6 months group when compared to WKY (**a**,**b**,**e**,**f**), in the 12 months group was found a more significant presence of AQP4 in the SHR (**c**,**d**,**g**,**h**) rats. Stain intensities in relative units for AQP4 (**i**,**j**) are represented as the means ± SD (*n* = five animals per group). The differences between WKY and SHR were significant when applied to a One-way ANOVA test with post hoc analysis using the Tukey post hoc test (* *p* < 0.05). BV: blood vessel; GU: grey units. Scale bars 10 μm.

**Figure 4 ijms-23-12277-f004:**
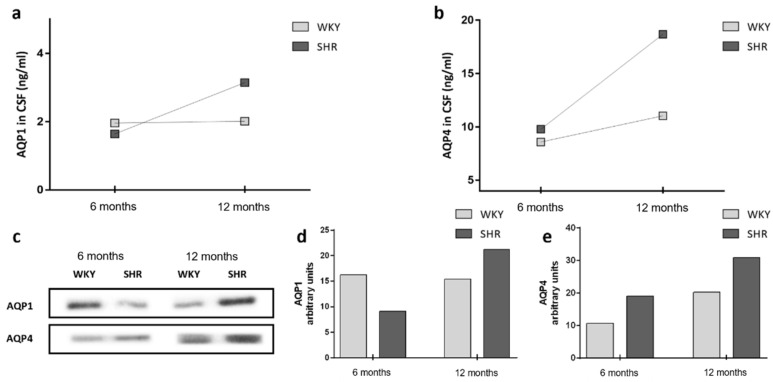
Variation in the timeline of AQP values in CSF. Representation of age-related mean values of AQP1 and AQP4 in CSF of WKY and SHR are shown in (**a**) and (**b**), respectively. Expression of AQP1 and AQP4 in CSF (**c**). Graphic representation of values of AQP1 and AQP4 in CSF by Western blot (**d**,**e**).

**Figure 5 ijms-23-12277-f005:**
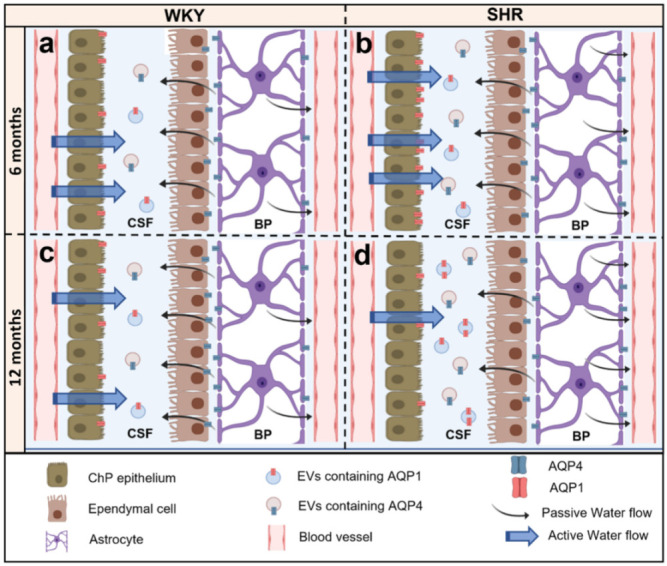
Schematic representation of the expression of AQP1 and AQP4 in 6 months and 12 months WKY and SHR rats. In 6 months and 12 months WKY (**a**,**c**), rat expression of AQP 1 is detected in the apical pole of the ChP epithelium associated with active production of CSF. It is also found in the CSF, most likely associated with EVs. Per contra, AQP4 is expressed in the astrocyte end-feet and the basolateral membrane of the ependymal cells associated with brain CSF homeostasis. In addition, it is found in the CSF. In 6 months SHR rats (**b**), AQP1 expression is significantly increased in the apical pole of the ChP epithelium, and AQP4 expression increases in the astrocytes end-feet but not in the ependymal cells. This AQP disbalance may contribute to the mechanisms that trigger ventriculomegaly. In the CSF, the expression of AQPs remains the same as in 6 months WKY. In the 12 months SHR rats (**d**), AQP1 expression significantly decreased in the ChP epithelium while proportionally increasing in the CSF compared to 12 months WKY (**c**). Thus, it could reduce CSF production as a possible compensatory mechanism for ventriculomegaly. In turn, AQP4 decreased in the ependymal layer while it increased in the CSF and in the astrocytes’ end-feet, which could also contribute to reducing the production of CSF (ependymal layer) and the increase of the CSF absorption (astrocyte end-feet).

**Table 1 ijms-23-12277-t001:** CSF levels (ng/mL) of AQP1 and AQP4 in 6 months and 12 months groups of WKY and SHR measured by ELISA expressed as mean ± SD.

Concentration (ng/mL)	Group	WKY	SHR
AQP1 in CSF	6 months	1.95 ± 0.13	1.64 ± 0.24
(mean ± SD)	12 months	2.01 ± 0.10	3.14 ± 0.80
AQP4 in CSF	6 months	8.59 ± 0.59	9.78 ± 0.82
(mean ± SD)	12 months	11.04 ± 0.39	18.68 ± 1.33

**Table 2 ijms-23-12277-t002:** Mean values of body weight, systolic blood pressure (SBP), and diastolic blood pressure (DBP) in the 6 months and 12 months groups of WKY and SHR rats. The SAP and DAP were measured by a tail-cuff method with the rats under conscious conditions using a noninvasive blood pressure measurement system. SBP and DBP were determined 3 times blind to the randomization sequence on each time point, and the mean values were used as the result. The same evaluator always took SAP and DAP measurements. The differences between WKY and SHR were significant when applied to a T-student test (* *p* < 0.05).

	Group	WKY	SHR
Body weight	6 months	386 ± 2.1	352 ± 1.9 *
(g. ± SD)	12 months	439 ± 2.7	398 ± 2.4 *
Systolic blood pressure	6 months	131 ± 2.9	179 ± 1.7 *
(mmHg ± SD)	12 months	125 ± 1.6	185 ± 0.8 *
Diastolic blood pressure	6 months	58 ± 1.7	64 ± 2.7 *
(mmHg ± SD)	12 months	68 ± 3.0	80 ± 2.1 *

## Data Availability

The datasets during and/or analyzed during the current study are available from the corresponding author on reasonable request.

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
