# Peer review of "Altered Expression of AQP1 and AQP4 in Brain Barriers and Cerebrospinal Fluid May Affect Cerebral Water Balance during Chronic Hypertension"

_ijms, 2022, doi:10.3390/ijms232012277_

Round 1
Reviewer 1 Report
This is a very nice study of SHR and control rats looking at the difference in AQ 1 and 4 expression both histologically and in CSF. The combination of the two methods is particularly important with the newer data about exosomes. This article has the potential to be pivotal in the future studies of clinical studies where there are clear aberrations in CSF production/absorption.
Here are some comments/questions that might help the readership:
1. There is an incomplete sentence in line 185 which might be important. Not sure if there is a missing phrase at start of 190 or if it is just not capitalized.
2. Is there data either from your group or in the literature that AQ are found in exosomes?
3. Error bars on the graphs would be helpful.
4. Can you clarify (perhaps I missed it) or highlight the number of different rats that showed the changes?
5. For example, a table that shows that 9/10 SHR showed the immunohistochemical changes vs 5/10 for the control.
6. Can you comment on how your findings tie in with the latest glymphatic/dural lymphatics theories?
Reviewer 2 Report
The manuscript by González-Marrero et al describes the expression of aquaporins (1 and 4) in the brain of a rat model for hypertension in comparison to a wild type strain. While this is potentially interesting for the understanding of water homeostasis in the brain, there are some serious issues that need to be addressed in the manuscript as it is.
The first concern is the age of the animals that were chosen for investigation: A 6 mo old rat is adult but a 12 mo rat is also adult, somewhat older. The description ‘aged’ is therefore misleading since rats can grow as old as 36 or more months (or is there a difference in the SHR strain?). So, to refer to the 12 mo old rat as aging (lane 85) is simply not correct. The issue is especially concerning regarding the presented data (Fig.1) that AQP1 is already increased in the 6 mo old rats. Likewise, the data in Fig. 3 are supposed to show increased AQP4 expression in the SHR strain independent of age. It would be necessary to have a younger age group where AQP1 expression is still the same in SHR and control rats, as well as an older group that would fall in the category of aging. This is crucial, because, as the authors acknowledge themselves, AQP4 increase has been reported previously in the SHR rat, and supposedly the novel aspect of this manuscript is aging. Since the quantification of immunofluorescence is difficult anyway, the lack of other age groups renders the data less significant.
Why is there such a difference between the CSF data generated by Elisa and western blot? Although it is stated that results matched (ln 155) for both methods, the Western blot band for AQP1 of the adult group shown in Fig. 4c and quantification in d show quite a difference yet the Elisa data do not (table 1).
Fig.5 as a summarizing figure is interesting but unfortunate in its illustration and statements: First, the blue color for ChP cells is similar to the AQP4 color and the channels are depicted fairly small. Second, the arrows, supposedly showing CSF production, look the same in the ChP and ependyma as it is also stated in the text (ln 266). However, these are different mechanisms: While the ChP CSF production is an active process across a barrier, the interstitial fluid flow across the ependyma is not (no barrier, no active process). This should be corrected or indicated. Third, the astrocytic endfeet AQP4 is referred as ‘associated with CSF reabsorption mechanisms’ but this is not at all clear in the brain, where it is also assumed to play a role in K+ syphoning.
The significance of the occurrence of AQPs in the CSF (in EVs or not) should be discussed more. Is this of relevance or a biproduct of other processes. Why is there an increase of AQP4 in the CSF when the ependyma expresses less?
Minor points in writing:
Abstract
Please indicate the abbreviation WKY (ln 20)
Ln 24 the wording ‘ CSF expression’ is misleading since there is no cellular expression there, you are simply measuring the presence of these proteins - please rephrase.
Introduction
Ln 40-42: the description is unprecise and misleading: there is no AQP expression in a barrier, cells in the neurovascular unit (astrocytes) and choroid plexus cells express it.
Ln 49, astrocytic endfeet face the pial basal lamina, not the subarachnoid.
Results
In the text, please give the units of the measures (even if arbitrary or grey values), the numbers alone are meaningless.
In Fig. 1, please indicate the age groups in the figures
Ln 89 ’when as’ is bad grammar
Ln 105 spelling ‘with’
Ln 130/131 ‘in the aged group’ is twice in the sentence
In the table 1, the lower lines are AQP4 data but it reads AQP1
Ln 183 has 2x ‘reduced’
Round 2
Reviewer 2 Report
The authors have adressed all my concerns. They revised the description of the two age groups clarifying that aged or old rats have not been Of course considering and comparing only these two groups (and not including a juvenile and old stage) renders the manuscript less novel. Nevertheless, I acknowlege the improvement compared to previous version of the manuscript.